# Corporate Social Responsibility and Sustainability: From a Corporate Governance Perspective

Lijuan Wu [1] and Shanyue Jin [2,*]

1   School of Accounting, Zibo Vocational Institute, Zibo 255300, China
2   College of Business, Gachon University, Seongnam 13120, Republic of Korea
*   Correspondence: jsyrena0923@gachon.ac.kr

**Abstract:** Sustainable corporate development has become essential for many enterprises in the context of economic globalization and fierce technological competition. In fact, it is being tackled at a strategic level by most companies. The fulfillment of corporate social responsibility (CSR) is significant in building a corporate image, improving brand competitiveness, and promoting sustainable corporate development. Simultaneously, the level of corporate governance is a crucial factor in an enterprise's long-term development. Therefore, this study clarifies whether CSR has a positive impact on the sustainable development of enterprises through empirical analysis; it also analyzes the effects of internal governance factors on the relationship between the two, from the perspective of corporate governance. A fixed-effects regression analysis was conducted on a sample of Chinese A-share listed companies from 2015 to 2019. According to the results, active CSR can promote sustainable development. Furthermore, corporate governance factors such as internal control, management capabilities, and accounting information quality have a moderating role in the CSR process on sustainable corporate development. This study provides a theoretical basis for future research on CSR and sustainable development, and its findings can inspire governments and enterprises from the perspective of corporate governance.

**Keywords:** corporate social responsibility; sustainable development; internal control; management capability; accounting information quality

## 1. Introduction

Global economic recovery is expected to remain weak for a long time, due to slow economic growth and a profound restructuring of the economy from incremental expansion to stock adjustment, with enterprises having no option but to optimize this gradual transition [1]. With changes in the macroeconomic environment and the continuous adjustment of the industrial structure, problems related to the sustainable development of enterprises, such as environmental pollution, ecological destruction, food safety, labor disputes, low production efficiency, and so on [2], have become increasingly prominent, which shows that many listed companies have a weak sense of social responsibility. In recent years, several large companies have collapsed, owing to a lack of social responsibility [3]. Poor internal control has resulted in huge losses and even bankruptcy in some cases.

Today, social responsibility and sustainable development are attracting a great deal of attention from all sectors of society, with active implementation of social responsibility becoming an inevitable choice for companies seeking sustainable development [4]. Furthermore, corporate social responsibility (CSR) is no longer about "if" but about "how" [5]. How to fulfill CSR to promote sustainable development is a question that needs to be answered by every company. Therefore, CSR is crucial for achieving sustainable development [6,7]. CSR implies the need for companies to protect and enhance the present and future welfare of society and organizations through various business and social actions, and to guarantee just and sustainable benefits to multiple stakeholders [8]. Most of the

existing literature studies the sustainable development of enterprises from the perspective of entrepreneurship and macroeconomic factors, but few consider the impact of social responsibility. Therefore, it is of great practical significance to explore whether corporate social responsibility will promote the sustainable development of enterprises.

In modern corporate governance, introducing CSR significantly enhances the integrity and effectiveness of internal controls [9]. Internal control, as an essential part of corporate governance, creates a favorable environment and provides the necessary safeguards for the implementation of CSR. Simultaneously, by actively fulfilling their CSR, enterprises can gain the trust and support and enhance the resources of their stakeholders, as well as improve their reputation and social influence and contribute to sustainable development [10].

The ownership and management rights of an enterprise are separate. Management plays an important role in an enterprise's daily operations and activities, as well as its future development. Based on the explicit factors of neoclassical economic theories, Jensen (1993) first found that executive power can significantly impact business performance [11]. However, neoclassical economic theories assume that management is homogeneous. Studies in later years introduced broad characteristic variables, measured in terms of age and education, and replaced management heterogeneity with demographic characteristics, thereby addressing this issue. It was not until Aghion et al. (2001) proposed the management improvement hypothesis that it was pointed out that the competencies of management teams differ across firms, which could affect the effectiveness of management decisions to some extent and alter the firm's prospects [12].

Simultaneously, corporate and academic communities consider the quality of accounting information as an essential basis for management to make business decisions and investors to select target companies for investments. As governments continue to improve their accounting standard systems and increase supervision of the capital market, the quality of accounting information has continuously improved and has nearly been perfected. Regarding the economic consequences of accounting information quality, previous studies have found that it can reduce enterprises' equity capital costs [13,14], improve the efficiency of corporate investment [15,16], and optimize capital allocation efficiency [17], among other benefits.

To sum up, few pieces of literature consider the impact of social responsibility on the sustainable development ability of enterprises, and even fewer comprehensively examine the moderating effect of corporate governance on the relationship between social responsibility and sustainable development. In a complex economic environment, it is of great significance to study the relationship between corporate social responsibility and sustainable development, as well as the moderating effects of internal control, management capabilities, and accounting information quality on social responsibility and sustainable development from corporate governance. The study will clarify the mechanism between them, enrich the literature on social responsibility's economic consequences, and promote enterprises' sustainable and stable development.

The contributions of this study are as follows. First, it empirically verifies the impact of CSR on corporate sustainable development and broadens the research area of CSR. Second, it examines the moderating effects that internal control, management capabilities, and accounting information quality have on the relationship between CSR and corporate sustainable development, providing a theoretical basis for promoting corporate governance. Third, it explores the impact mechanism and realization path of corporate sustainable development, providing a theoretical basis for promoting the sustainable development of listed companies.

The structure of this study is arranged as follows. Section 2 offers the literature review and hypotheses. Section 3 presents the research design of this paper, including sample selection, definition of the variables, and model design. In Section 4, the empirical results are presented, reporting the main test and robustness tests. The conclusions and implications of this study are discussed in Section 5.

## 2. Literature Review and Hypotheses

The theory of corporate sustainability, though it emerged relatively recently, has developed rather quickly. The theory points out that it is difficult for companies to adapt to a rapidly changing social environment. With many businesses failing, it is becoming increasingly essential for companies to gain momentum despite the crisis [18]. Corporate sustainability is not only about speed but also about quality of growth [19,20]. The theory of corporate sustainable development advocates energy conservation, environmental protection, and improved operational efficiency, and requires corporate development to be in harmony with resources and the carrying capacity of the environment, matching the needs of society and human development.

### 2.1. Corporate Social Responsibility and Sustainable Development

Based on stakeholder theory, CSR is a multidimensional measurement concept [21,22]. From a new institutional economics perspective, CSR is the monitoring and constraint of profit-seeking behavior by stakeholders in a market economy situation. In addition to considering its business situation, a business must also consider its impact on society and the natural environment, and fulfill its responsibilities and obligations to its stakeholders (shareholders, employees, consumers, partners, government, and the public) [23,24]. The impact of CSR on sustainable development is mainly reflected in two aspects. First, actively fulfilling CSR is an effective way to enhance the soft power of enterprises and establish a favorable social image. Pavla Vrabcová et al. (2021) find that fulfilling CSR is a process of accumulating and integrating human, social, and other resources, which is conducive to enhancing the sustainable competitiveness of enterprises [25]. Lopez Belen et al. (2022) argued that corporate development should not only focus on short-term profit maximization but also on CSR, as it plays an increasingly important role in promoting corporate success and social progress [26]. Second, companies can gain an early advantage with regard to sustainable development through social responsibility performance [27]. Doukas John A. et al. (2021) argued that in building and developing corporate culture, the integration of CSR should be fully considered to make it unique and effective [28]. The efficient integration of the two relationships can bring about innovation in culture, products, and processes, and enable enterprises to obtain unique competitive advantages, thereby enhancing economic returns. According to Samet Marwa et al. (2022), social responsibility is not only an inevitable act of responding to the times, but also an essential driver of competitiveness [29]. As a critical strategic resource, it can help companies maintain long-term relationships, conducive to maintaining sustainable competitive advantage in the long run. This shows that CSR is a "win-win strategy", which can help introduce innovative ways and provide an early advantage in terms of sustainable development. In summary, the following hypothesis is proposed.

**Hypothesis 1.** *CSR contributes positively to corporate sustainable development.*

### 2.2. The Moderating Effect of Internal Control

According to the principal–agent theory, the objectives of the principal and agent are not aligned. There is information asymmetry; therefore, it is impossible to expect the agent to act in full accordance with the risk preferences of the principal, and the principal can only take measures to maximize the interests of both parties, resulting in agency costs [30,31]. The essence of internal control is the principal–agent problem, which must be resolved to maintain a balance between the interests of all parties in the firm, reduce transaction costs, mitigate agency conflicts, and reduce opportunistic behavior of the management, thereby improving the quality of corporate surplus [32]. As a necessary institutional arrangement for enterprises, internal control plays an active role in the interplay between CSR and sustainable development. For all enterprise activities, internal control plays a process monitoring and institutional guidance role. The better the quality of internal control, the more compliant is the enterprise's behavior in fulfilling its social responsibility [33,34].

As a corporate activity, CSR is inevitably monitored and guided by internal controls at the implementation stage [35], thus affecting the performance output of CSR. A sound internal control system can play a major role in optimizing and controlling the implementation of CSR, thereby ensuring that the activities have been conducted in an orderly and efficient manner, and thus improving the competitiveness of enterprises [36]. When, in the process of fulfilling social responsibility, companies are subjected to internal control, they can conduct effective social responsibility risk management, reduce CSR risks, and promote the achievement of their strategic goals [37]. A sound internal control system can coordinate and optimize the path of information exchange between enterprises and governments, thereby improving the transparency of information between them. This can facilitate governments' understanding of CSR compliance and make it easier for enterprises to obtain government support, thus helping them achieve long-term development. In fulfilling their social responsibility, enterprises can help prevent CSR risks and promote sustainable development by standardizing and improving their internal control systems and operations [38]. Therefore, the more robust the internal control system of the enterprise, the more significant the role of CSR in promoting sustainable development. Therefore, this study also proposes the following hypothesis:

**Hypothesis 2.** *An improved internal control system of enterprises entails a greater role of corporate social responsibility in promoting their sustainable development.*

### 2.3. The Moderating Effect of Management Capacity

Hambrick and Mason (1984) proposed the upper echelon theory, which argues that a firm's management cannot have a comprehensive understanding of all aspects, and that the management's cognitive structure and values determine their understanding and judgement of information [39]. Management traits influence decision making and affect corporate performance [40]. As the central force in corporate decision making, the management is directly responsible for all production, operation, investment, and financial decisions, as well as development strategies. They are also responsible for fulfilling CSR, the implementation of which has a long-term economic impact and is strongly constrained by management capacity [41]. Therefore, management capacity will significantly impact CSR performance and thus affect the enterprises' long-term development [42]. Some previous studies have proved that CSR performance is in line with management's internal decision making, primarily determined by management's capacity [43]. The management actively performs social responsibility, which positively promotes enterprise development [44]. Some scholars have argued that competent managers can better align corporate investment decisions with CSR strategies, which helps to manage resources effectively, reduce costs, maximize the benefits of CSR, and promote sustainable corporate development [45]. Simultaneously, management capability can effectively reduce the degree of information asymmetry [46]. The more robust the management capability is, the more effective its decision making will be, and therefore, information released will be easily recognized by the outside world, better balancing the interests among stakeholders and promoting the fulfillment of CSR and sustainable development of the enterprise. Based on the above analysis, this study proposes the following hypothesis:

**Hypothesis 3.** *A more robust management capacity entails a more significant role of CSR in promoting corporate sustainable development.*

### 2.4. The Moderating Effects of Accounting Information Quality

The theory of information asymmetry refers to the idea that some individuals in an economy have access to information that others do not, and that the degree of asymmetry is influenced by the behavior of firms and individuals [47]. Information asymmetry and the principal–agent problem caused by it generate huge transaction costs for enterprises, creating obstacles in fulfilling social responsibilities and improving their sustainable development ability [48]. Accounting information is an essential part of the information

provided by enterprises to the outside world and its governance mechanisms. This is the most common way to transmit information to enterprises' capital markets. The disclosure behavior and contents of enterprise accounting information play an essential role in signal transmission. At the same time, according to the upper echelon theory, managements influence and even determine accounting behaviors and policies such as corporate disclosure and surplus management [49]. Numerous studies confirm the influence of management on the quality of accounting information [50,51]. Selectively disclosing self-serving information and manipulating data and their presentation in financial reports often undermine the reliability of information, thereby influencing users' perceptions of the company's performance and growth prospects. Thus, though low-quality accounting information can benefit management, it increases the agency costs of the company. CSR can reduce the need for vigilance and the concern of stakeholders in terms of surplus management and indemnify them against surplus management that cannot be sustained in the long term.

Amiran M. et al. (2022) conducted an empirical analysis of listed companies in different countries: the poorer the quality of accounting information, the higher the degree of corporate manipulation of surplus management [52]. They argued that the higher the degree of surplus management, the more actively a company fulfills its social responsibility. This is because fulfillment of social responsibility by an enterprise can improve its social image, and stakeholders will not be suspicious of its accounting information, thus providing room for sustainable development. The quality of accounting information has been shown to positively contribute to enterprises' sustainability [53]. The social responsibility performance of enterprises also forms a specific substitution effect. The quality of accounting information weakens the positive impact of CSR on enterprises' sustainable development abilities [54]. Burke Q. et al. (2020) conducted an in-depth study on the motivation of enterprises to fulfill social responsibility [55]. This research clearly shows that enterprises whose accounting information is less authentic and reliable often actively perform their social responsibilities to make external stakeholders or auditors believe in the financial data and increase the credibility of accounting information, thus promoting the sustainable development of enterprises. Therefore, last but not least, this study also proposes the following hypothesis:

**Hypothesis 4.** *Poorer quality of accounting information entails a more significant contribution of CSR to corporate sustainable development.*

## 3. Research Design

### 3.1. Data and Samples

The sample consisted of Chinese A-share listed companies from 2015 to 2019. Data on internal control were obtained from the internal control index of listed companies, published by Shenzhen Dibo Enterprise Risk Management Technology Co. Ltd. (DIB) (Shenzhen, China). Data on social responsibility were obtained from the CSR rating scores in Hexun's social responsibility measurement system, and other data were obtained from the WIND and the China Stock Market & Accounting Research (CSMAR) database (https://www.wind.com.cn/, accessed on 25 October 2021; https://www.gtarsc.com/, accessed on 25 October 2021). The following treatments were applied to the data: companies in the ST and *ST sectors, and those in the financial sector were excluded. To eliminate the influence of outliers on the regression results, we Winsorized all continuous variables at the 1% and 99% levels.

### 3.2. Definition of the Variables

#### 3.2.1. Dependent Variable

Corporate sustainability was chosen as the dependent variable in this study. Corporate sustainability is a firm's ability to continue to grow profitably and robustly in its existing competitive field and future business development environment, in the pursuit of survival and growth. This study draws on Liao et al. (2022) and uses Van Horne's static model

to measure corporate sustainability [56]. This model was chosen to study the sustainable development of a firm in terms of its profitability and competitiveness.

### 3.2.2. Independent Variables

This study selected CSR as an independent variable. Previous studies have developed various evaluation measures for CSR; however, no unified standard has been established. Hexun (Beingjing, China), as an international authoritative, independent third-party rating agency, has been committed to providing the public with the most scientific and professional social responsibility evaluation information. After the release of annual and social responsibility reports, Hexun.com (accessed on 25 October 2021) uses the expert scoring method to comprehensively evaluate the social responsibility of enterprises from multiple perspectives. From the stakeholder perspective, different weights are set for each stakeholder's social responsibility from each of the five dimensions. The higher the score, the better the enterprise's social responsibility performance. This study adopted the social responsibility score released by Hexun to measure the fulfillment of CSR, which is also the current method adopted by many previous studies [57].

### 3.2.3. Moderating Variables
#### Internal Control

The internal control index, based on the five internal control objectives, is the first indicator system released by the DIB to measure the quality of internal control of all listed companies in China. The release of the index marks an era of quantifiable assessment of these indicators, and fills a research gap in the quantitative measurement of internal controls in China. The index system fully reflects the actual situation of internal control in China, dividing the five objectives of the internal control index system into three levels: basic, operational, and strategic. Therefore, considering the authority and completeness of the data, this study followed Li X. (2022) and selected the DIB internal control index to rate the level of internal control of listed companies in China [58].

#### Management Capability

Management capability refers to the ability to manage various resources of the business effectively. Firms with higher management capabilities can achieve higher output for a given level of production factor input. This study draws on Dermerjian et al.'s study (2012) to account for management capabilities using data envelopment analysis (DEA) and building a corresponding model, assuming that both firm and management capability influence a firm's efficiency [59].

In the model, operating cost, selling and administrative expenses, net fixed assets, intangible assets, goodwill, and research and development expenditure were selected as input variables. Operating revenue was selected as the output variable, and production efficiency was estimated as follows:

$$\text{Max}_v \theta = \text{Sales}/(v_1\text{Cost} + v_2\text{SG\&A} + v_3\text{PPE} + v_4\text{Intangible} + v_5\text{Goodwill} + v_6\text{R\&D}) \quad (1)$$

The production efficiency calculated in the above formula ranges from 0 to 1 and includes the influence of enterprise level and management capabilities. Measuring management capability directly by size leads to overestimated risk. Based on this, a Tobit regression model was constructed. Assets, market share, free cash flow, age, and business complexity were selected as factors influencing enterprise production efficiency. Excluding the influence of enterprise level, the residual of the regression model is managers' ability.

$$\theta = \alpha_1 + \alpha_2\text{InAssets} + \alpha_3\text{Ms} + \alpha_4\text{Fcf} + \alpha_5\text{Age} + \alpha_6\text{BHHI} + \sum \text{Year}_i + \varepsilon_i \quad (2)$$

#### Quality of Accounting Information

The better a listed company's surplus management, the worse the quality of accounting information. In this study, the modified Jones model [60] was used to calculate the surplus

management level of controllable accrued profits to measure the quality of accounting information. The larger the manipulable accrued profits, the greater the propensity for surplus management and the poorer the quality of accounting information. The formulas are shown in Equations (3) and (4).

$$\frac{TA_{i,t}}{A_{i,t-1}} = \alpha + \beta_1\left(\frac{1}{A_{i,t-1}}\right) + \beta_2\left(\frac{\Delta REV_{i,t}}{A_{i,t-1}}\right) + \beta_3\left(\frac{PPE_{i,t}}{A_{i,t-1}}\right) + \varepsilon_{i,t} \tag{3}$$

$$DA = \frac{TA_{i,t}}{A_{i,t-1}} - \left[\beta_1\left(\frac{1}{A_{i,t-1}}\right) + \beta_2\left(\frac{\Delta REV_{i,t} - \Delta REC_{i,t}}{A_{i,t-1}}\right) + \beta_3\left(\frac{PPE_{i,t}}{A_{i,t-1}}\right)\right] \tag{4}$$

$TA_{i,t}$ represents the total accrued profit of company $i$ in period $t$, the calculation method is operating profit minus net cash flow from operating activities; $A_{i,t-1}$ represents the total assets of company $i$ at the end of period $t-1$; $\Delta REV_{i,t}$ is the increase in revenue from the main business of company $i$ in period $t$; $\Delta REC_{i,t}$ is the increase in the book value of accounts receivable for company $i$ in period $t$; and $PPE_{i,t}$ is the book value of fixed assets of company $i$ at the end of period $t$. This study first regresses Equation (3) to obtain the estimated values of the coefficients $\beta_1$, $\beta_2$, $\beta_3$ and the residual absolute value $\varepsilon$, inputting $\beta_1$, $\beta_2$, $\beta_3$ into Equation (4) to calculate and take the absolute value to obtain DA.

### 3.2.4. Control Variables

In order to exclude other factors from interfering with the research results and to combine the findings of previous studies [45,61], the control variables selected in this study were enterprise size (SIZE), asset–liability ratio (LEV), firm growth (Growth), executive shareholding ratio (MH), independent director ratio (IDR), ownership concentration (Top1), year (Year), and industry (Ind). Table 1 presents the variables.

**Table 1.** Variables definition.

| Variables Type | Variable Name | Variable Symbol | Meaning and Description |
|---|---|---|---|
| Dependent variable | Corporate sustainability | SGR | Net profit margin on sales × total asset turnover × income retention rate × equity multiplier/(1—net profit margin on sales × total asset turnover × income retention rate × equity multiplier) |
| Independent variables | Corporate social responsibility | CSR | Hexun scoring |
| Adjustment variables | Internal control | IC | Dib Index |
| | Management capability | MA | Calculated using Equations (1) and (2) |
| | Quality of accounting information | DA | Calculated using Equations (3) and (4) |
| Control variables | Enterprise size | SIZE | Natural logarithm of total assets at the end of the year |
| | Asset–liability ratio | LEV | Total liabilities/total assets |
| | Firm growth | GROWTH | Operating income growth rate |
| | Executive shareholding ratio | MH | Value of shares held by executives/total share capital |
| | Independent director ratio | IDR | Number of independent directors/number of board of directors |
| | Ownership concentration | Top1 | Percentage of shareholding of the largest shareholder |
| | Year | Year | Annual dummy variables |
| | Industries | Ind | Industry dummy variables |

### 3.3. The Model Design

To test Hypothesis 1, that CSR has a positive contribution to corporate sustainability, fixed-effects regression Model (5) was constructed [62,63].

$$\text{SGR} = \beta_0 + \beta_1\text{CSR} + \beta_2\text{SIZE} + \beta_3\text{LEV} + \beta_4\text{GROWTH} + \beta_5\text{MH} + \beta_6\text{IDR} \\ + \beta_7\text{TOP1} + \sum\text{YEAR} + \sum\text{Ind} + \varepsilon \tag{5}$$

If $\beta_1$ in model (5) is greater than 0 and significant, then the better the CSR fulfillment, the more sustainable the company.

To test Hypothesis 2, which states that the better the internal control, the more significant the contribution of CSR to corporate sustainability, Model (6) was constructed.

$$\text{SGR} = \beta_0 + \beta_1\text{CSR} + \beta_2\text{ICQ} + \beta_3\text{CSR} \times \text{ICQ} + \beta_4\text{SIZE} + \beta_5\text{LEV} \\ + \beta_6\text{GROWTH} + \beta_7\text{MH} + \beta_8\text{IDR} + \beta_9\text{TOP1} \\ + \sum\text{YEAR} + \sum\text{Ind} + \varepsilon \tag{6}$$

If $\beta_3$ is positive and significant in Model (6), it indicates that internal control has a positive moderating effect on CSR and sustainability.

To test Hypothesis 3, which states that the better the management capability, the more significant the contribution of CSR to corporate sustainability, Model (7) was constructed.

$$\text{SGR} = \beta_0 + \beta_1\text{CSR} + \beta_2\text{MA} + \beta_3\text{CSR} \times \text{MA} + \beta_4\text{SIZE} + \beta_5\text{LEV} \\ + \beta_6\text{GROWTH} + \beta_7\text{MH} + \beta_8\text{IDR} + \beta_9\text{TOP1} \\ + \sum\text{YEAR} + \sum\text{Ind} + \varepsilon \tag{7}$$

If $\beta_3$ in Model (7) is positive and significant, it indicates that management capability has a positive moderating effect on CSR and sustainability.

To test Hypothesis 4, namely, the moderating effect of accounting information quality on CSR and corporate sustainability, Model (8) was constructed.

$$\text{SGR} = \beta_0 + \beta_1\text{CSR} + \beta_2\text{DA} + \beta_3\text{CSR} \times \text{DA} + \beta_4\text{SIZE} + \beta_5\text{LEV} \\ + \beta_6\text{GROWTH} + \beta_7\text{MH} + \beta_8\text{IDR} + \beta_9\text{TOP1} \\ + \sum\text{YEAR} + \sum\text{Ind} + \varepsilon \tag{8}$$

In Model (8), the moderating variable DA is the quality of accounting information. In this study, the modified Jones model was used to calculate the manipulable accrued profit, with larger values indicating poorer accounting information quality and smaller values indicating better quality. If $\beta_3$ in Model (8) is positive and significant, it indicates that the poorer the quality of accounting information, the more significant the contribution of CSR to the sustainable development of the enterprise.

## 4. Empirical Results

### 4.1. Descriptive Statistics

Table 2 presents the results of the descriptive statistics of the variables. The table shows that the maximum value of corporate sustainability is 0.3375, the minimum value is −0.0230, the mean value is 0.0669, and the standard deviation is 0.0596, indicating that the overall sustainability of the sample companies is poor and varies significantly between different companies. The maximum value of CSR is 68.6000, the minimum value is 3.4600, the mean value is 23.4503, and the standard deviation is 10.4551, indicating that the sample companies' overall fulfillment of social responsibility is poor.

**Table 2.** Descriptive statistics.

| Variable | Obs | Mean | Std. Dev. | Min | Max |
|---|---|---|---|---|---|
| SGR | 9861 | 0.0669 | 0.0596 | −0.0230 | 0.3375 |
| CSR | 9861 | 23.4503 | 10.4551 | 3.4600 | 68.6000 |
| IC | 9861 | 6.4826 | 0.1227 | 5.7809 | 6.6948 |
| MA | 9861 | 0.2214 | 0.2913 | −0.2910 | 0.7432 |
| DA | 9861 | 0.0531 | 0.0530 | 0.0006 | 0.2731 |
| SIZE | 9861 | 22.3771 | 1.3062 | 20.1154 | 26.1858 |
| LEV | 9861 | 0.4075 | 0.1913 | 0.0625 | 0.8541 |
| GROWTH | 9861 | 0.2050 | 0.3757 | −0.4229 | 2.4012 |
| MH | 9861 | 0.1087 | 0.1696 | 0.0000 | 0.6446 |
| IDR | 9861 | 0.3789 | 0.0651 | 0.2500 | 0.6000 |
| Top1 | 9861 | 34.1504 | 14.4784 | 8.5378 | 73.5623 |

### 4.2. Correlation Analysis

Table 3 lists the correlation coefficients of the variables. The correlation coefficient between CSR and sustainable development is 0.3294 and significant at the 1% level, indicating a positive relationship between CSR and sustainable development. The mean value of VIF for the regression model is less than 10, and the maximum VIF value is 1.84, which allows us to ignore the impact of multicollinearity on the main results of this study.

**Table 3.** Correlation analysis.

| Variables | SGR | CSR | IC | MA | DA | SIZE | LEV | GROWTH | MH | IDR | Top1 |
|---|---|---|---|---|---|---|---|---|---|---|---|
| SGR | 1 | | | | | | | | | | |
| CSR | 0.3294 *** | 1 | | | | | | | | | |
| IC | 0.2518 *** | 0.2406 *** | 1 | | | | | | | | |
| MA | 0.0869 *** | 0.0691 *** | 0.0622 *** | 1 | | | | | | | |
| DA | 0.0013 | −0.0553 *** | −0.0358 *** | −0.0327 *** | 1 | | | | | | |
| SIZE | 0.1238 *** | 0.1914 *** | 0.1241 *** | 0.1450 *** | −0.0484 *** | 1 | | | | | |
| LEV | 0.1099 *** | −0.0127 | 0.0334 *** | 0.0820 *** | 0.0322 *** | 0.5756 *** | 1 | | | | |
| GROWTH | 0.2034 *** | 0.0476 *** | 0.1414 *** | 0.0187 * | 0.0611 *** | 0.0357 *** | 0.0714 *** | 1 | | | |
| MH | 0.0298 *** | −0.0318 *** | 0.0318 *** | −0.0412 *** | 0.0123 | −0.4015 *** | −0.2677 *** | 0.0576 *** | 1 | | |
| IDR | −0.0015 | −0.0023 | 0.0303 *** | −0.0093 | 0.0283 *** | −0.0643 *** | −0.0573 *** | −0.0036 | 0.1257 *** | 1 | |
| Top1 | 0.0551 *** | 0.1085 *** | 0.0740 *** | 0.0369 *** | −0.0205 ** | 0.1895 *** | 0.0848 *** | −0.0301 *** | −0.0964 *** | 0.0268 *** | 1 |

Note: *, **, *** represent the significance levels of 1%, 5%, and 10%, respectively.

### 4.3. Regression Analysis

Column (1) of Table 4 shows the regression results of Model (5). The coefficient of CSR is 0.0020 and is significantly correlated at the 1% level, indicating a significantly positive relationship between CSR and corporate sustainable development, which means that listed companies can enhance their sustainable development ability by fulfilling CSR. Therefore, Hypothesis 1 is supported.

**Table 4.** Regression analysis.

| Variables | (1) SGR | (2) SGR | (3) SGR | (4) SGR |
|---|---|---|---|---|
| CSR | 0.0020 *** (36.0473) | −0.0372 *** (−14.5425) | 0.0014 *** (20.3518) | 0.0016 *** (20.5148) |
| IC | | −0.0481 *** (−5.3429) | | |
| c.CSR#c.IC | | 0.0060 *** (15.2696) | | |
| MA | | | −0.0517 *** (−11.4314) | |
| c.CSR#c.MA | | | 0.0025 *** (14.9274) | |
| DA | | | | −0.1473 *** (−6.3583) |

**Table 4.** *Cont.*

| Variables | (1) SGR | (2) SGR | (3) SGR | (4) SGR |
|---|---|---|---|---|
| c.CSR#c.DA | | | | 0.0082 *** (8.0583) |
| SIZE | 0.0009 (1.6278) | 0.0000 (0.0211) | 0.0011 * (1.9051) | 0.0012 ** (2.0128) |
| LEV | 0.0401 *** (10.7871) | 0.0383 *** (10.5173) | 0.0372 *** (10.1159) | 0.0388 *** (10.4462) |
| GROWTH | 0.0287 *** (19.4279) | 0.0249 *** (17.0855) | 0.0279 *** (19.1327) | 0.0281 *** (19.0457) |
| MH | 0.0185 *** (5.1704) | 0.0161 *** (4.6084) | 0.0185 *** (5.2255) | 0.0186 *** (5.2219) |
| IDR | −0.0035 (−0.4092) | −0.0077 (−0.9362) | −0.0035 (−0.4241) | −0.0040 (−0.4732) |
| Top1 | 0.0001 ** (2.5498) | 0.0001 ** (1.9629) | 0.0001 *** (2.9058) | 0.0001 *** (2.7004) |
| Constant | −0.0280 ** (−2.2570) | 0.3144 *** (5.3234) | −0.0179 (−1.4602) | −0.0243 * (−1.9506) |
| Year | Control | Control | Control | Control |
| Industries | Control | Control | Control | Control |
| Observations | 9861 | 9861 | 9861 | 9861 |
| Adjusted r2 | 0.1826 | 0.2199 | 0.2027 | 0.1881 |
| F | 297.1698 *** | 294.6611 *** | 264.7869 *** | 240.2810 *** |

Note: t-values in parentheses; *, **, *** represent the significance levels of 1%, 5%, and 10%, respectively.

Columns (2) and (3) of Table 4 show the results of the test of Models (6) and (7). The interaction term coefficient $\beta_3$ between CSR and internal control and management capability is greater than 0 and is significant at the 1% level, indicating that internal control and management capability can have a positive moderating effect on CSR and corporate sustainable development. Column (4) of Table 4 shows the test results of Model (8). The interaction term coefficient $\beta_3$ between CSR and accounting information quality is greater than 0 and significant at the 1% level, indicating that the poorer the quality of accounting information, the more significant the contribution of CSR to corporate sustainable development. This supports Hypotheses 2–4.

### 4.4. Robustness Tests

To verify the reliability of the findings, this study has drawn from existing research [64], a replacement measure of corporate sustainability. SGR2 = subsist–profit ratio × net profit margin on sales × total asset turnover × equity multiplier. The regression analysis of the model, consistent with the primary model, verified the robustness and reliability of the results obtained in this study, and the Table 5 shows the results of the robustness test.

**Table 5.** Robustness test results.

| Variables | (1) SGR2 | (2) SGR2 | (3) SGR2 | (4) SGR2 |
|---|---|---|---|---|
| CSR | 0.0019 *** (36.2311) | −0.0308 *** (−13.2335) | 0.0013 *** (21.2344) | 0.0015 *** (20.5066) |
| IC | | −0.0311 *** (−3.7960) | | |

**Table 5.** *Cont.*

| Variables | (1) SGR2 | (2) SGR2 | (3) SGR2 | (4) SGR2 |
|---|---|---|---|---|
| c.CSR#c.IC | | 0.0050 *** (13.9600) | | |
| MA | | | −0.0422 *** (−10.2534) | |
| c.CSR#c.MA | | | 0.0021 *** (13.5985) | |
| DA | | | | −0.1411 *** (−6.7009) |
| c.CSR#c.DA | | | | 0.0076 *** (8.2279) |
| SIZE | 0.0013 ** (2.4048) | 0.0004 (0.7517) | 0.0014 *** (2.6184) | 0.0014 *** (2.7698) |
| LEV | 0.0350 *** (10.3497) | 0.0335 *** (10.1289) | 0.0326 *** (9.7215) | 0.0338 *** (10.0194) |
| GROWTH | 0.0263 *** (19.5790) | 0.0227 *** (17.1650) | 0.0256 *** (19.2896) | 0.0258 *** (19.2168) |
| MH | 0.0189 *** (5.8171) | 0.0165 *** (5.1884) | 0.0189 *** (5.8640) | 0.0190 *** (5.8680) |
| IDR | −0.0039 (−0.5081) | −0.0079 (−1.0524) | −0.0040 (−0.5215) | −0.0043 (−0.5638) |
| Top1 | 0.0001 *** (2.6946) | 0.0001 ** (2.0668) | 0.0001 *** (3.0149) | 0.0001 *** (2.8497) |
| Constant | −0.0293 *** (−2.5998) | 0.2009 *** (3.7396) | −0.0206 * (−1.8430) | −0.0253 ** (−2.2368) |
| Year | Control | Control | Control | Year |
| Industries | Control | Control | Control | Industries |
| Observations | 9861 | 9861 | 9861 | 9861 |
| Adjusted r2 | 0.1861 | 0.2223 | 0.2031 | 0.1916 |
| F | 303.4050 *** | 298.1097 *** | 264.5273 *** | 245.3504 *** |

Note: t-values in parentheses; *, **, *** represent the significance levels of 1%, 5%, and 10%, respectively.

## 5. Conclusions and Implications

In recent years, China has increased support for enterprises to fulfill their social responsibilities and vigorously advocated sustainable development. In this context, we are discussing corporate social responsibility's impact on enterprises' sustainable development. It can not only expand the research on the factors affecting the sustainable development of enterprises but also provide a reference for the decision making of listed companies. Chinese A-share listed companies from 2015 to 2019 were considered as the sample. We empirically analyzed the relationship between CSR and corporate sustainability using a fixed-effects regression model. At the same time, we examined the role of internal corporate governance in the relationship between social responsibility and sustainable development of enterprises. We tested its moderating role in the relationship between social responsibility and sustainable development of enterprises from three aspects: internal control, management capability, and accounting information quality. The following conclusions were drawn. (1) CSR has a positive impact on enterprise sustainability. The fulfillment of CSR is conducive to enhancing corporate sustainable development ability. (2) Internal control and management capabilities positively moderate the process of social responsibility, influencing the corporate sustainable development ability. The better the internal control

and management capability, the greater the promotion effect of CSR on the corporate sustainable development ability. (3) The quality of accounting information plays a moderating role in enterprises fulfilling their social responsibilities, and affects their sustainable development abilities. The poorer the quality of accounting information, the more significant the role of CSR in promoting the sustainable development of enterprises). Furthermore, we conducted an in-depth study on the relationship between CSR and corporate sustainable development ability, revealing the mechanism of CSR affecting corporate sustainable development ability and demonstrating the moderating role of internal control, management capabilities, and accounting information quality in the impact of CSR on corporate sustainable development ability from the perspective of internal corporate governance.

This paper puts forward the theoretical and managerial implications based on the above research and analysis. First, in realizing sustainable development, enterprises must take the initiative to fulfill their social responsibilities, improve their corporate system, and formulate sustainable development strategies. Enterprises should implement social responsibility in daily management and corporate strategy. Enterprises should actively improve the internal governance of the company. The management should pay attention to establishing a correct view of internal control, strengthen the importance of internal control and scientific understanding, and improve the construction and operation of the internal control system. Enterprises should pay more attention to the capabilities assessment of the management and adopt a reasonable compensation incentive mechanism linked to the performance of CSR to promote the management to make more favorable decisions for the performance of CSR. At the same time, if the quality of accounting information is poor, enterprises should attach more importance to the role of CSR in promoting sustainable development. Second, government departments should actively guide enterprises to fulfill social responsibilities. The government can implement policies and actions such as government subsidies and tax incentives for enterprises that pay attention to fulfilling social responsibilities, guide and cultivate enterprises to form a good habit of fulfilling social responsibilities, strengthen the formulation of corporate social responsibility laws and regulations, increase the intensity and cost of punishment for violations, and give full play to the guiding role of the law. We will improve the review and evaluation of CSR performance via external rating agencies, strengthen the attention and supervision of media on CSR performance, and constantly improve the external environment required for fulfilling CSR.

This study has some limitations as well. Besides the factors considered in this study, many other factors also affect the sustainable development ability of enterprises, such as innovation ability and asset quality; however, this study has not considered them. These variables can be used for further analyses in future studies.

**Author Contributions:** Data curation and draft, L.W.; methodology, review, and editing, S.J. All authors have read and agreed to the published version of the manuscript.

**Funding:** This research received no external funding.

**Institutional Review Board Statement:** Not applicable.

**Informed Consent Statement:** Not applicable.

**Data Availability Statement:** Not applicable.

**Conflicts of Interest:** The authors declare no conflict of interest.

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
