# Peer review of "Corporate Social Responsibility and Sustainability: From a Corporate Governance Perspective"

_sustainability, doi:10.3390/su142215457_

Round 1

Reviewer 1 Report

Thank you for giving me the opportunity to review this interesting paper.

In order to investigate the relationship between CSR and corporate sustainability this article uses a fixed-effects regression model. It examined the role of internal corporate governance mechanisms in the relationship between social responsibility and sustainable development of firms.
Overall, I find this research very interesting and well done.
Here are some suggestions that I hope will be useful to the authors to improve the manuscript:
1.      Introduction section: my suggestion is to better emphasize literature gap to support the research conducted. Moreover, my suggestion is to explain the research questions to which you intend to answer with the analysis conducted in order to make the reader more aware of what the goals of the work are and, at the same time, make an easier connection to those who the main contributions of the work are.
2.      Method: I suggest to the authors to improve the methodology section enhancing with references to studies that were done using the same technique.
3.      Conclusion section: I suggest to the authors to better empathize the theoretical and managerial implications, even with regards to policy makers.

I hope these comments will prove helpful to the authors.

Author Response

Thank you for giving me the opportunity to review this interesting paper.

In order to investigate the relationship between CSR and corporate sustainability this article uses a fixed-effects regression model. It examined the role of internal corporate governance mechanisms in the relationship between social responsibility and sustainable development of firms.
Overall, I find this research very interesting and well done.
Here are some suggestions that I hope will be useful to the authors to improve the manuscript:
【1-1】. Introduction section: my suggestion is to better emphasize literature gap to support the research conducted. Moreover, my suggestion is to explain the research questions to which you intend to answer with the analysis conducted in order to make the reader more aware of what the goals of the work are and, at the same time, make an easier connection to those who the main contributions of the work are.

Reply: We have added the contents as follows: (P. 2)

Most of the existing literature studies the sustainable development of enterprises from the perspective of entrepreneurship and macroeconomic factors, but few consider the impact of social responsibility. Therefore, it is of great practical significance to deeply explore whether corporate social responsibility will promote the sustainable development of enterprises.

To sum up, few pieces of literature consider the impact of social responsibility on the sustainable development ability of enterprises, and even fewer comprehensively examine the moderating effect of corporate governance on the relationship between social responsibility and sustainable development. In a complex economic environment, it is of great significance to study the relationship between corporate social responsibility and sustainable development, as well as the moderating effects of internal control, management capabilities, and accounting information quality on social responsibility and sustainable development from corporate governance. The study will clarify the mechanism between them, enrich the literature on social responsibility's economic consequences, and promote enterprises' sustainable and stable development.

【1-2】. Method: I suggest to the authors to improve the methodology section enhancing with references to studies that were done using the same technique.

Reply: We have added the references as follows: (P. 8)

To test hypothesis 1, that CSR has a positive contribution to corporate sustainability, a fixed-effects regression Model [62, 63] (5) was constructed.

【1-3】.Conclusion section: I suggest to the authors to better empathize the theoretical and managerial implications, even with regards to policy makers.

Reply: We have added the theoretical and managerial implications as follows: (P. 13)

This paper puts forward the theoretical and managerial implications based on the above research and analysis. First, in realizing sustainable development, enterprises must take the initiative to fulfill their social responsibilities, improve their corporate system, and formulate sustainable development strategies. Enterprises should implement social responsibility into daily management and corporate strategy. Enterprises should actively improve the internal governance of the company. The management should pay attention to establishing a correct view of internal control, strengthen the importance of internal control and scientific understanding, and improve the construction and operation of the internal control system. Enterprises should pay more attention to the capabilities assessment of the management and adopt a reasonable compensation incentive mechanism linked to the performance of CSR to promote the management to make more favorable decisions for the performance of CSR. At the same time, if the quality of accounting information is poor, enterprises should attach more importance to the role of CSR in promoting sustainable development. Second, Government departments should actively guide enterprises to fulfill social responsibilities. The government can implement policies and actions such as government subsidies and tax incentives for enterprises that pay attention to fulfilling social responsibilities. Guide and cultivate enterprises to form a good habit of fulfilling social responsibilities. Third, strengthen the formulation of corporate social responsibility laws and regulations, increase the intensity and cost of punishment for violations, and give full play to the guiding role of the law. We will improve the review and evaluation of CSR performance by external rating agencies, strengthen the attention and supervision of media on CSR performance, and constantly improve the external environment required for fulfilling CSR.

I hope these comments will prove helpful to the authors.

Reviewer 2 Report

Dear author(s), Thank you very much for submitting your valuable work to sustainability. I read this article. This study has a lot of contribution to “Corporate Social Responsibility and Sustainability: From a Corporate Governance Perspective”. However, in some lines and sentences there still minor issues.  I am requesting the authors to have a relook and correct it.

1.     Considering the topic: Very interesting. Please provide thorough explanation of the connections between the paper´s topic and the topic of sustainability. You should start in the introduction, continue in the theoretical background and you should address this link also within the discussion (or conclusions).

2.     Despite the extensive literature review, there is a lack of current references related to the objects under study regarding the recent use of Corporate Sustainability Performance, the authors should provide more updated references.

https://doi.org/10.1016/j.ribaf.2022.101732

https://doi.org/10.1016/j.jebo.2021.09.005

https://doi.org/10.1016/j.eneco.2021.105431

3.     Considering “Data analysis”, my suggestion is to provide clear explanation about hypothesis acceptation of rejection already here. In discussion you could provide additional explanations. 

4.     Please also correct the some citations in the last section. Tables are also not according to the journal standard. You may correct the table now or later if accepted.  

Author Response

Dear author(s), Thank you very much for submitting your valuable work to sustainability. I read this article. This study has a lot of contribution to “Corporate Social Responsibility and Sustainability: From a Corporate Governance Perspective”. However, in some lines and sentences there still minor issues.  I am requesting the authors to have a relook and correct it.

【2-1】.     Considering the topic: Very interesting. Please provide thorough explanation of the connections between the paper´s topic and the topic of sustainability. You should start in the introduction, continue in the theoretical background and you should address this link also within the discussion (or conclusions).

 Reply: We have added the contents as follows: (p.12)

In recent years, China has increased support for enterprises to fulfill their social responsibilities and vigorously advocated sustainable development. In this context, we are discussing corporate social responsibility's impact on enterprises' sustainable development. It can not only expand the research on the factors affecting the sustainable development of enterprises but also provide a reference for the decision-making of listed companies.

【2-2】.     Despite the extensive literature review, there is a lack of current references related to the objects under study regarding the recent use of Corporate Sustainability Performance, the authors should provide more updated references.

https://doi.org/10.1016/j.ribaf.2022.101732

https://doi.org/10.1016/j.jebo.2021.09.005

https://doi.org/10.1016/j.eneco.2021.105431

 Reply: We have added the references as follows:

Therefore, CSR is crucial for achieving sustainable development [6, 7]. (p.1)

Corporate sustainability is not only about speed but also about quality of growth [19, 20]. (p.3)

Lopez Belen et al.(2022) argues that corporate development should not only focus on short-term profit maximization but also on CSR, as it plays an increasingly important role in promoting corporate success and social progress [26]. (p.3)

Samet Marwa et al. (2022), social responsibility is not only an inevitable act of responding to the times, but also an essential driver of competitiveness [29]. (p.3)

Better the quality of internal control, more compliant is the enterprise’s behavior in fulfilling its social responsibility [33, 34]. (p.3)

Numerous studies confirm the influence of management on the quality of accounting information [50, 51]. (p.4)

【2-3】.     Considering “Data analysis”, my suggestion is to provide clear explanation about hypothesis acceptation of rejection already here. In discussion you could provide additional explanations.

Reply: We have added the contents as follows: (p. 13)

Furthermore, this paper conducts an in-depth study on the relationship between CSR and corporate sustainable development ability, reveals the mechanism of CSR affecting corporate sustainable development ability, and demonstrates the moderating role of internal control, management capabilities, and accounting information quality in the impact of CSR on corporate sustainable development ability from the perspective of internal corporate governance.

【2-4】.     Please also correct the some citations in the last section. Tables are also not according to the journal standard. You may correct the table now or later if accepted.  

Reply: We will correct the table later if accepted.

Reviewer 3 Report

The abstract, introduction, and literature review are written in a thoughtful and balanced manner. It was suggested that the introduction refers to the structure and development of the work.

Author Response

【3-1】The abstract, introduction, and literature review are written in a thoughtful and balanced manner. It was suggested that the introduction refers to the structure and development of the work.

Reply: We have added the contents as follows: (p.2)

The structure of this study is arranged as follows: Section 2 offers the literature review and hypotheses. Section 3 presents the research design of this paper, including sample selection, definition of the variables, and model design. In section 4, the empirical results are presented, reporting the main test and robustness tests. The conclusions and implications of this study is discussed in section 5.